# Intercropping Forage Sorghum with Sunnhemp at Different Seeding Rates to Improve Forage Production

Haley M. Mosqueda [1,*], Brock C. Blaser [1], Susan A. O'Shaughnessy [2] and Marty B. Rhoades [1]

1 Department of Agricultural Sciences West Texas A&M University, Canyon, TX 79016, USA; bblaser@wtamu.edu (B.C.B.); mrhoades@wtamu.edu (M.B.R.)
2 USDA-ARS Conservation and Production Research Laboratory, Bushland, TX 79012, USA; susan.oshaughnessy@usda.gov
* Correspondence: haley.mosqueda@ndsu.edu

**Abstract:** Forage sorghum (*Sorghum bicolor* (L.) Moench) is well established in the Texas High Plains as a drought-tolerant forage that often requires additional feed to provide adequate protein for livestock. Intercropping sunnhemp (*Crotalaria juncea* L.), a legume, with forage sorghum, may increase crude protein. However, the optimal intercrop seeding ratio of sunnhemp to sorghum to improve crude protein content and maintain sufficient biomass is unknown. A two-year field experiment was conducted near Canyon, TX, USA, in 2020 and 2021 using sunnhemp intercropped at three seeding rates (16.8, 33.6, and 50.4 kg ha$^{-1}$) with forage sorghum at four seeding rates (0, 2.8, 5.6, and 11.2 kg ha$^{-1}$) under drip irrigation. This study was conducted to (1) evaluate growth potential for sunnhemp in a semiarid environment, (2) find a seeding ratio that can maintain forage sorghum dry matter production and improve forage quality, and (3) determine if a midseason harvest can be supported and further improve quality of the forage produced. Midseason, full-season, regrowth biomass, and forage quality were evaluated. Results indicated that a sunnhemp–forage-sorghum intercrop produced dry matter comparable to forage sorghum when sufficient heat units were obtained in the growing season. Forage with higher nutritive value was produced when the intercrop was harvested twice.

**Keywords:** forage sorghum; legumes; intercrops





## 1. Introduction

The Texas High Plains (THP) region has a semiarid climate and is primarily known for grain and forage crop production. Texas is the top producing state for sorghum (*Sorghum bicolor* (L.) Moench) silage at 25% of the total production in the U.S. [1]. When grown for livestock, additional feed is typically added since forage sorghum and other forage grasses provide a limited amount of crude protein (CP) and other nutrients necessary in the diets of cattle [2]. Agriculture is threatened in the THP due to the continuing depletion of the Ogallala Aquifer [3,4], which is the main source of water for agriculture and the surrounding municipalities. In this region, the recharge rate is approximately 0.06 cm per year, while the depletion rate could be greater than 45 m [5]. Due to the limited water supply and competition for water among sectors, it is important to determine companion crops that provide optimal nutrition to cattle and biomass yield and are drought- and heat-tolerant due to the need to improve irrigation water use efficiency (IWUE) in crops grown in the region [6].

Studies in the literature reported varied benefits of intercropping. In general, mixed cropping can provide weed [7] and pest [8] suppression, increased yield [9], and positive long-term effects on soil quality [10]. Irrigation water use efficiency can be improved by establishing an intercropping system that reduces evaporation from the soil [11]. Abdel-Wahab et al., 2019 [12] reported that intercropping legumes with cereal species improved the total agro-ecosystem due to the contributions of N from the legumes and

balancing fiber and crude protein concentration in forages. Liu et al., 2023 [13] reported that legumes intercropped with cereals did not change the nutritive value of feed but did improve the effective use of the land. More specifically, the intercropping of forage sorghum with forage legumes can increase crude protein (CP) [14]. The choice of crops used in a mixed crop system is crucial to provide the desired outcome.

Sunnhemp is a warm-season legume that has been grown since the beginning of mechanized food production as a cover crop, soil-improving green manure, fiber, and forage for goats in India [15]. Due to its high protein content (leaves 25–30% and stems 8–10%) and potential grazing capabilities, sunnhemp could be an ideal forage for livestock [16]. As a legume, nitrogen (N) production from this crop has the potential to provide enough N for itself, a companion crop, and contribute to soil health when used in multi-year production [16,17]. When grown in the southeastern U.S., sunnhemp biomass levels reached 14 Mg ha$^{-1}$ after 120 d after planting [18], Bhardwaj et al., 2005 [19] reported similar yields at 13.4 Mg ha$^{-1}$ for sunnhemp grown in the mid-Atlantic region of the U.S., and Cook et al., 1998 [20] reported biomass yields of 16.2 Mg ha$^{-1}$ for sunnhemp grown in the Rio Valley of Texas (humid climate). Long taproots and well-developed lateral roots can pull water from lower depths in the soil profile [15]. This, along with its ability to maintain growth with little water once established, may support the arising needs for producers in the Texas High Plains to produce livestock feed with less irrigation.

There are no current herbicides registered for use in sunnhemp crops; however, studies have shown that its rapid growth generally suppresses the growth of weeds [15,21,22]. Higher plant density of sunnhemp suppresses weeds earlier in the season [21]. As sunnhemp growth progresses through the season, the biomass of weeds may be reduced up to 50% [22], suggesting that rapid growth and the height of this species directly influence weed reduction.

Additionally, the rapid growth and potential height for this crop may serve as a benefit for intercropping with a rapidly growing, tall crop such as forage sorghum. Choosing a legume that grows taller than common legumes may increase DM production in a sorghum intercrop and potentially avoid competition for light with the sorghum. Shorter, bushy legumes such as cowpea (*Vigna unguiculata* (L.) Walp.) or soybean (*Glycine max* L.) are commonly intercropped with forage sorghum to improve forage quality [14]. However, these legumes often do not contribute to DM accumulation when intercropped with sorghum, which is likely due to the significant impact of the sorghum leaf area index and reducing available light to the legumes below the sorghum canopy [23].

The drought tolerance of sorghum makes it an ideal crop to grow in the High Plains region of Texas. Forage sorghum heights can reach 1.5–5 m depending on the variety used [24]. Bean et al., 2013 [25] reported that forage sorghum yields averaged 14.2 Mg ha$^{-1}$ in Bushland, Texas, when total available water averaged 830 mm and thermal energy ranged from 1500 to 1690 growing degree days (GDDs).

The root system of sorghum is fibrous and can grow to soil depths of 2.4 m [26]. A study was conducted by Blum and Arkin, 1984 [27] in Temple, TX, on the root length and root length density of sorghum grown under irrigated and water-stressed conditions. The root growth of late-maturing sorghum occurred primarily laterally in the 0.9–1.2 m soil depth 60 days after emergence. Root length density increased up to 1.5 m in irrigated sorghum. Water-stressed sorghum root length increased to 1.8 m, but root length density did not increase past the 1.2 m depth.

Limited research on sunnhemp has been conducted in a semiarid environment, and minimal known research has been published on sunnhemp intercropped with forage sorghum for cattle forage in the Texas High Plains. The objectives of this study were to (1) evaluate the potential for sunnhemp growth in a semiarid environment, (2) find a seeding ratio that can maintain forage sorghum dry matter production and improve forage quality, and (3) determine whether a midseason forage harvest can be supported and further improve the quality of the forage produced.

## 2. Materials and Methods

### 2.1. Environmental Conditions

The experiment was conducted at the West Texas A&M University Nance Ranch (34.9704° N, 101.803° W, elevation 1097 m) near Canyon, TX, USA, on an Olton Clay loam (fine, mixed, superactive, thermic Aridic Paleustolls), a well-drained, moderately slowly permeable soil [28]. Semiarid growing conditions in Canyon, TX, USA, include a 30-year average of up to 508 mm of rainfall per year and an average temperature of 31.1 °C during the summer growing season, according to U.S. Climate Data. Growing degree days (GDDs) were calculated as follows:

$$GDD = [(T_{max} + T_{min})/2] - T_{base},$$

where $T_{max}$ and $T_{min}$ were daily maximum and minimum recorded temperatures, respectively (Table 1). The base temperature ($T_{base}$) for the growth of sunnhemp used in the calculation was 10 °C [17]. No cutoff temperature was used for sunnhemp GDD calculations based on the methods used by Balkcom et al., 2011 [17] and Lepcha et al., 2018 [29]. Forage sorghum GDDs were calculated using $T_{base}$ = 10 °C and a cutoff temperature = 30 °C, whereby $T_{max}$ was limited to 30 °C for GDD calculations, respectively.

**Table 1.** Climatic conditions for 2020 and 2021 growing seasons near Canyon, TX, USA.

| Month | Average Air Temp | Total Rainfall | Forage Sorghum GDDs [a] | Sunnhemp GDDs |
|---|---|---|---|---|
| | °C | mm | °C d | °C d |
| **2020 Growing Season** | | | | |
| June | 25 | 20 | 254 | 291 |
| July | 28 | 43 | 464 | 553 |
| August | 26 | 21 | 431 | 509 |
| September | 20 | 8 | 144 | 144 |
| Total | | 92 | 1293 | 1497 |
| **2021 Growing Season** | | | | |
| June | 24 | 83 | 221 | 252 |
| July | 24 | 68 | 424 | 452 |
| August | 25 | 7 | 431 | 479 |
| September | 23 | 78 | 144 | 239 |
| Total | | 236 | 1358 | 1422 |

[a] GDDs, growing degree days.

### 2.2. Agronomics and Experimental Design

The field was prepared by making two tillage passes with a rotary tiller to loosen the soil after three years of fallow following sorghum and pearl millet (*Pennisetum glaucum* (L.) Leeke) in 2020 and two years of fallow following sesame (*Sesamum indicum* L.) in 2021. A third-party laboratory analysis of soil nutrients was performed prior to planting in both years. Sufficient P and K were found in the soil to support sorghum growth; however, N was lacking. Granular nitrogen fertilizer (urea) was applied across the field at a rate of 196 kg N ha$^{-1}$ based on recommendations from a commercial laboratory after analyzing soil samples taken prior to planting. This was done to ensure sufficient N for the forage sorghum and to reduce any variation in sunnhemp monocrop treatments as the intercrops would have additional N that may increase sunnhemp biomass accumulation. The fertilizer was applied using a hand-held broadcast fertilizer spreader on 11 June 2020 and 14 June 2021. The fertilizer was tilled into the soil using a rotary tiller for a third pass over the field. Planting occurred on 12 June 2020 and 14 June 2021.

The study was arranged as a split-plot nested design with twelve treatments and four replications over two consecutive years, 2020 and 2021. The split plot was implemented with two harvests, where half of each plot was harvested at 45 and 90 DAP, and

the other half was only harvested at 90 DAP. Each of the 48 experimental plots measured 18.6 m$^2$ (3.05 m × 6.10 m) which were then split in half to implement a midseason harvest treatment on 27 July 2020 and 29 July 2021. North or south half of the plot was randomly chosen to hand harvest samples for the midseason harvest. The sunnhemp (vns, Petcher Seed Company, Fruitdale, AL) was inoculated with 3.0 g kg$^{-1}$ seed of a cowpea-type, peat-based *Bradyrhizobium* immediately before planting. Sorghum seeds were drilled 3.8 cm deep at seeding rates of 0, 2.8, 5.6, and 11.2 kg ha$^{-1}$ (designated as forage sorghum treatments: no, lo, md, hi) in rows spaced 76 cm apart, each intercropped with sunnhemp seed drilled at seeding rates of 16.8, 33.6, and 50.4 kg ha$^{-1}$ (designated as sunnhemp treatments: lo, md, hi). The sunnhemp was planted between sorghum rows and spaced 19 cm apart, with three rows of sunnhemp between each row of sorghum. Each intercropped plot contained four rows of sorghum and nine rows of sunnhemp (Figure 1). In sunnhemp monocrop treatments, the sunnhemp seeds were planted in rows spaced 19 cm apart with thirteen rows of sunnhemp per plot. Sorghum was drilled in rows 19 cm apart along the perimeter of the field to create a 1.5 m border around the plots.

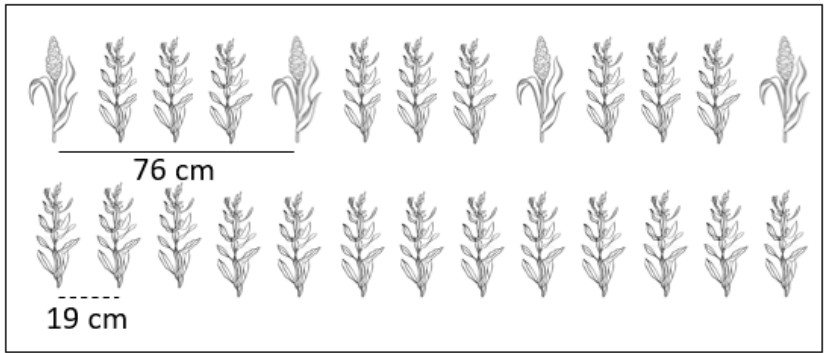

**Figure 1.** Intercropped sorghum and sunnhemp (**top**) where four sorghum rows were spaced 76 cm apart with three sunnhemp rows in between spaced 19 cm apart. Sunnhemp containing no sorghum (**bottom**) had thirteen rows spaced 19 cm apart.

All plots were irrigated uniformly using a flow-metered surface drip line system designed similar to that used by Machicek et al., 2019 [30] and Crookston et al., 2020 [31]. Each experimental unit contained two drip lines spaced 152 cm apart with emitters spaced every 61 cm. In 2020, a total of 605 mm of irrigation water was applied, and the area received 92 mm of precipitation, according to the weather station located 100 m from the field. In 2021, a total of 324 mm of irrigation was applied, and 304 mm precipitation was received. The field was sprayed with Zeta-Cypermethrin (0.35%) at 31 mL·L$^{-1}$ of water on 19 July 2020 to combat an infestation of gray blister beetles (*Epicauta fabricii*). The same chemical was applied at the same rate on 27 July 2021 to reduce the grasshopper (*Melanoplus differentialis* (Thomas)) population that began to strip the leaves from the sunnhemp. Weed pressure was minimal in all treatments as the growing season progressed; therefore, methods of weed control were not used.

### 2.3. Canopy Development and Irrigation Water Use Efficiency

Photosynthetic active radiation (PAR) was measured bi-weekly using an AccuPAR LP-80 Ceptometer (Decagon Devices, Pullman, WA, USA) starting at 25 DAP. Measurements were obtained by taking one above-canopy (ambient) reading and two below-canopy readings from the ceptometer which was placed horizontally to the soil surface and perpendicular across five plant rows. Measurements were collected under full-sun conditions between 1100 and 1400 h. Percent light interception was calculated using the following equation:

$$[\mu(PAR_{b1} + PAR_{b2})/PAR_a] \times 100 - 100,$$

where $PAR_{b1}$ and $PAR_{b2}$ are each below-canopy PAR reading, and $PAR_a$ is the above-canopy PAR reading. Not subtracting 100 from the total would result in the percent light penetration into the canopy.

Irrigation water use efficiency (IWUE) was calculated at the end of each growing season using the following equation:

$$IWUE = Y_g/IRR,$$

where $Y_g$ is the biomass yield (g m$^{-2}$), and IRR is the irrigation water applied (mm).

### 2.4. Dry Matter and Forage Quality

At midseason, 45 DAP, a small sickle was used to hand cut plants in each half of every plot down to a height of 30 cm (27 July 2020 and 29 July 2021) to determine midseason dry matter (DM) production and forage quality (FQ) and to provide an evaluation of regrowth potential of the intercrop. At the time of cutting, sorghum plants were approaching 1.5 m in height at the V8 to V9 stage. Sunnhemp plants were approaching 1.0 m in height in the vegetative stage at the time of cutting. Two biomass samples from two, 1 m$^2$ quadrats of each half-plot were collected and dried in an oven for a minimum of 72 h at 60 °C. Two rows of sorghum with the three rows of sunnhemp in between were collected. In sunnhemp plots containing no sorghum, five rows of sunnhemp were collected. Each species was separated at the time of cutting and drying to determine the biomass ratio of each species in the intercrop. After drying and weighing, each species from the same experimental unit was combined to represent the full treatment for further analysis. The samples were ground using a woodchipper and sent to ServiTech Laboratories in Amarillo, TX, USA for a relative feed value test to determine CP, acid detergent fiber (ADF), neutral detergent fiber (NDF), total digestible nutrients (TDN), and relative feed value index (RFV).

At the end of the growing season, 90 DAP, two biomass samples from two, 1 m$^2$ quadrats of each half-plot were cut 5 cm above soil level using the same methods as the midseason harvest to evaluate FQ from the full season and regrowth. Forage samples were again analyzed by ServiTech Laboratories.

At ServiTech, the samples were ground through a Wiley mill (Arthur H. Thomas Co., Philadelphia, PA, USA) to pass through a 1 mm screen and analyzed using wet chemistry. Crude protein was determined using the combustion method, the AOAC official method 990.03 (AOAC International, 2012). Percent ADF and NDF were determined using Ankom technology methods 5 and 6 (Ankom, 2006, Macedon, NY, USA), modified from AOAC official methods 973.18 and 2002.04, respectively. The calculation used to determine TDN was

$$TDN = (NFC \times 0.98) + (CP \times 0.87) + (FA \times 0.97 \times 2.25) + (NDFn \times NDFDp/100) - 10,$$

where NFC is non-fiber concentrate, CP is crude protein, FA is lipid content, NDFn is neutral detergent fiber, and NDFDp is neutral detergent fiber digestibility. The formula used to calculate RFV was

$$RFV = (DDM \times DMI)/1.29,$$

where digestible dry matter (DDM) = $88.9 - (0.779 \times \%ADF)$ and dry matter intake (DMI) = $120/\%NDF$. These calculations were performed by ServiTech; therefore, the values used in these calculations were not provided. These parameters were estimated based on the digestibility of full-bloom alfalfa hay fed to dairy cattle, which follow the USDA hay quality designation guidelines [13].

### 2.5. Statistical Analysis

Proc GLM was used to carry out a two-way analysis of variance (SAS Studio ver. 3.8, SAS Institute Inc., Cary, NC, USA 2020) where the first factor was sunnhemp seeding rates (3 levels), and the second factor was forage sorghum seeding rates (4 levels) for a

total of 12 possible treatment combinations. Year had a significant effect on DM, forage nutritive values, and IWUE. Therefore, the data were separated by year to determine the main effects of forage sorghum and sunnhemp seeding rates on forage production and quality and IWUE. All treatment interactions and differences were considered significant if $p \leq 0.05$. When analyzing DM in 2021, comparison of mean values was performed using Kruskal–Wallis since the DM data were not normally distributed.

## 3. Results and Discussion

### 3.1. Weather

Weather conditions during the 2020 growing season were more favorable for crop growth than during 2021 (Table 1). In 2020, the average air temperature was consistent with the 30-year average of air temperature recorded from 1989 to 2019 in Canyon, TX, USA. In 2021, the overall average air temperature was lower than the 30-year average. Total rainfall was 41% lower in 2020 and 137% higher in 2021 than the 30-year average. Sunnhemp GDDs accumulated more slowly in 2021 than in 2020. Sorghum GDD accumulation was similar in both years where the maximum temperature threshold used for forage sorghum did not account for the higher temperatures observed in the 2020 growing season compared to the 2021 growing season.

### 3.2. Canopy Development

The percent light interception from PAR increased from 25 to 36 DAP and leveled off to achieve maximum light interception at 48 DAP in 2020 (Figure 2a,b) just before midseason harvest. In 2021, light interception increased from 25 to 42 DAP to reach a maximum average by 50 DAP (Figure 2c,d), similar to the trend for light interception in 2020. For sunnhemp treatments with no forage sorghum, the average maximum light interception was less than 80% in 2021, nearly 20% lower than the average maximum light interception of 96% in treatments containing forage sorghum at all seeding rates. Darapuneni et al., 2018 [23] found that light interception was not significant between sorghum monocrop and sorghum–legume intercrops. In their study, plants reached maximum light interception at or after 60 DAP when the sorghum and legume species were planted at a 1:1 row ratio where rows were spaced 37.5 cm apart. A study conducted by Dzvene et al., 2023 [32] on maize and sunnhemp intercropping in semiarid South Africa evaluated planting time and stand density effect on light interception. Maize was sown in rows 200 cm apart with five rows of sunnhemp between maize rows, spaced 30 cm apart. When the maize and sunnhemp were planted simultaneously, maximum light interception was reached near 100 DAP. Based on these results, the earlier timing of maximum light interception reached in the present study may be explained by the closer planting arrangements of sorghum rows spaced 76 cm apart with sunnhemp rows planted in between spaced 15 cm apart. The denser canopy obtained by close row spacing may also have impacted differences in light interception found at different plant densities of each species in the intercrop.

Light interception in all midseason harvest treatment plots decreased by 37–45% immediately after harvest, then reached a maximum light interception of 85% by 75 DAP (Figure 2a,c). Maximum average light interception obtained from full-season harvested treatment plots was 99%. This is similar to results found by Machicek et al., 2019 [30] where light interception was evaluated after cutting sorghum sudangrass or pearl millet at 30, 45, and 90 DAP. Although the data in their study were analyzed by GDDs, similar trends occurred in the reduction and recovery of light interception in sorghum sudangrass cut at 45 and 90 DAP.

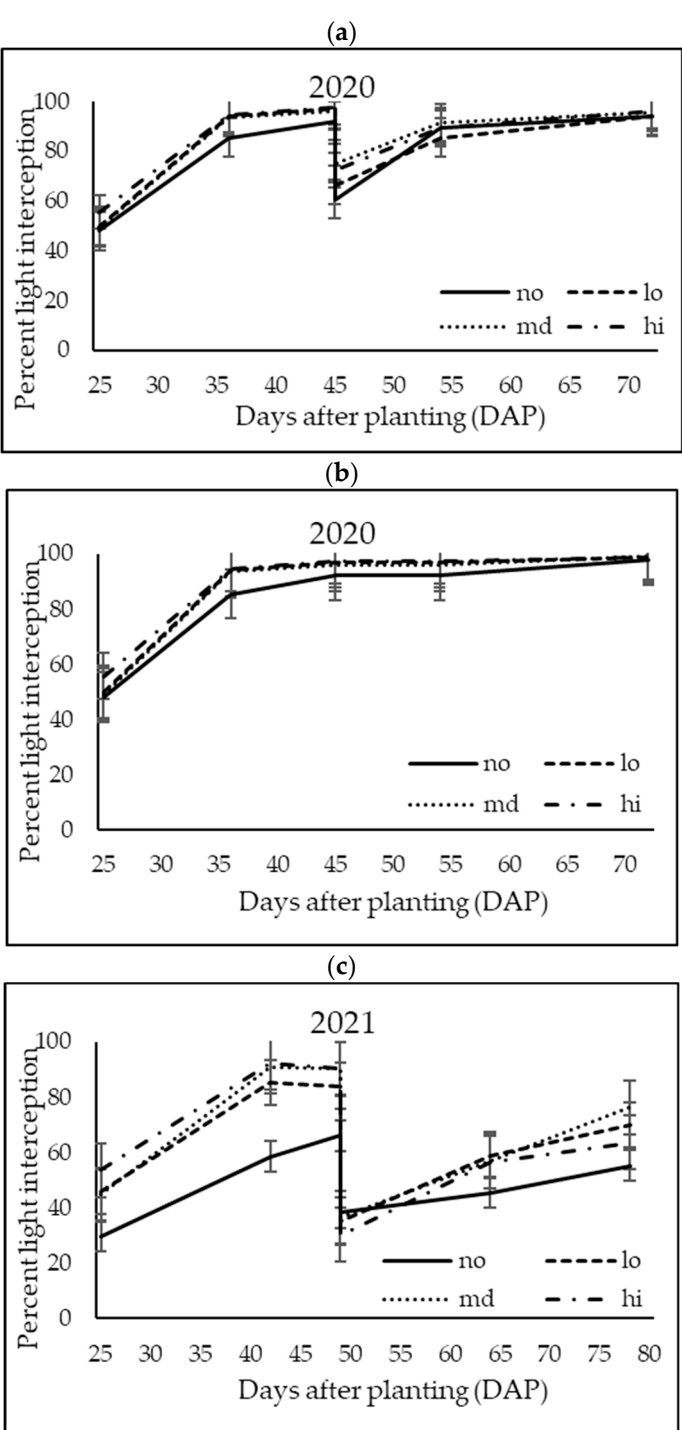

**Figure 2.** *Cont.*

(**d**)

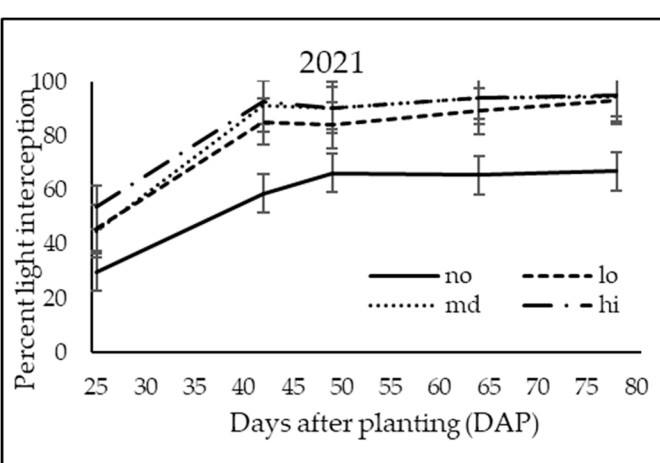

**Figure 2.** Mean percent light interception from photosynthetic active radiation (PAR) by sorghum seeding rate from midseason harvest treatments and regrowth in 2020 (**a**) and 2021 (**c**) and from full-season harvest treatments in 2020 (**b**) and 2021 (**d**). Error bars represent standard error for treatments at each timepoint.

*3.3. Irrigation Water Use Efficiency*

In both years, there was no sunnhemp seeding rate effect on IWUE. The average IWUE in treatments containing high and medium seeding rates of forage sorghum was not significantly different. In 2020, IWUE was greater in treatments with forage sorghum than in treatments with no forage sorghum with an average of 2.9 and 1.1 g m$^2$ mm$^{-1}$, respectively. In 2021, IWUE was again greater for FS-SH intercrop treatments than for the SH monocrop and greater for the md and hi FS plant density (5.6 kg ha$^{-1}$) than for the lo FS plant density (2.8 kg ha$^{-1}$) (Figure 3b). The average IWUE in treatments containing no forage sorghum was significantly less as compared with all other treatments at 0.6 g m$^2$ mm$^{-1}$, while the average IWUE in treatments containing low forage sorghum (3.9 g m$^2$ mm$^{-1}$) was different from the medium- and high-seeding-rate treatments (5.1 g m$^2$ mm$^{-1}$). These results are similar to those reported by O'Shaughnessy et al., 2023 [33] when sunnhemp was grown as a monocrop and an intercrop with sorghum. Sorghum is known as a drought-tolerant forage grown in the Texas High Plains [27], and therefore, it is expected to improve IWUE when grown in the area.

(**a**)

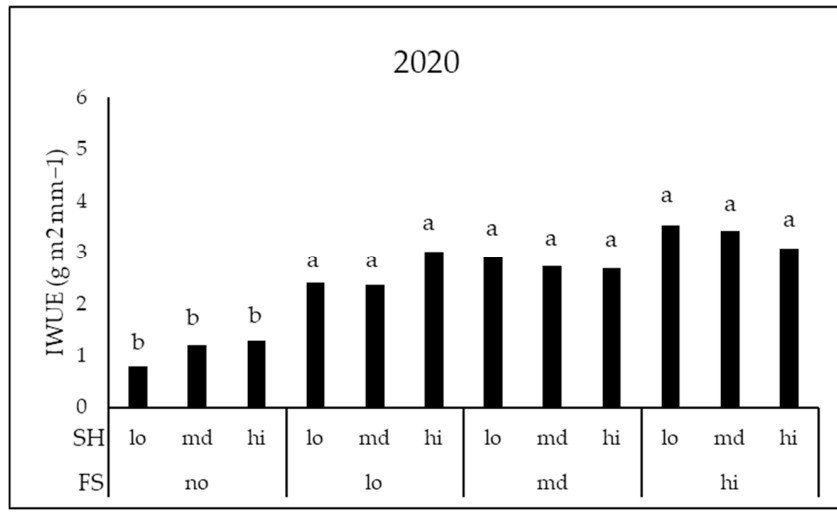

**Figure 3.** *Cont.*

**(b)**

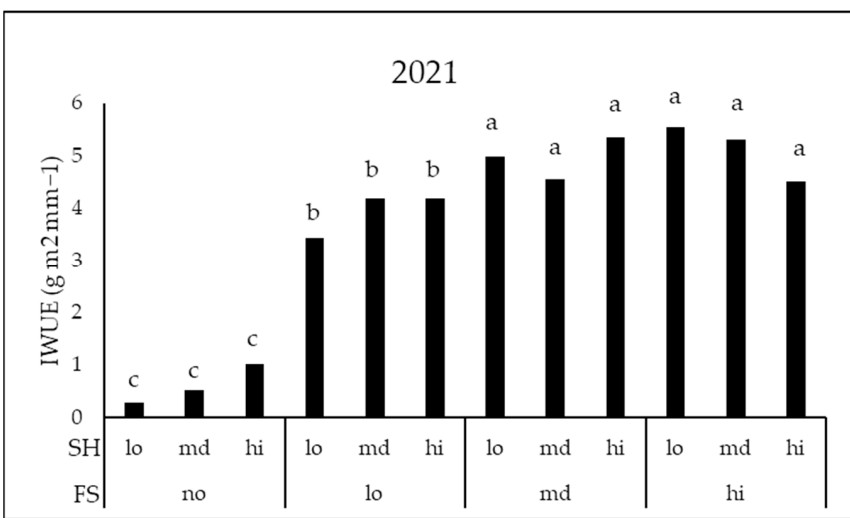

**Figure 3.** Irrigation water use efficiency (IWUE) of each intercrop seeding ratio in (**a**) 2020 and (**b**) 2021. Different letters within the same year are different at $p \leq 0.05$.

### 3.4. Dry Matter Production

Sunnhemp DM at full harvest did not change as the seeding rate of sunnhemp or forage sorghum increased but was higher in the monocrop SH treatments ($p < 0.001$) than in the FS-SH intercrop treatments (Figure 4). Forage sorghum DM increased as the seeding rate increased ($p < 0.001$). There was no forage sorghum X sunnhemp interaction on total intercrop dry matter production in 2020 ($p = 0.47$). Sunnhemp seeding rate had no effect on total crop DM in both years ($p = 0.96$ and $p = 0.88$). This is similar to the results obtained by Balkcom et al. [17] when comparing seeding rate effects on sunnhemp biomass and N production when planted as cover following wheat where no difference was reported for the biomass of sunnhemp planted at 17, 34, 50, and 67 kg ha$^{-1}$. Similar results were also reported in Italy where sunnhemp intercropped with sorghum biomass yields were 19 to 23 Mg ha$^{-1}$, and sorghum monocrop yields were 20 to 24 Mg ha$^{-1}$ [34]. The yields reported for monocrop sunnhemp were higher in their study, where 13 to 14 Mg ha$^{-1}$ was produced.

The seeding rate of forage sorghum influenced total crop DM in both years ($p < 0.001$). Mean dry matter at the no-FS treatment ($6.5 \pm 1.61$ in 2020 and $2.0 \pm 1.24$ in 2021) was significantly less than all other treatments ($p < 0.001$) (Figure 4a,b). Mean DM at the low forage sorghum seeding rate treatments ($15.5 \pm 1.98$ in 2020 and $12.7 \pm 1.37$) was significantly higher than the no-FS treatment ($p < 0.05$) and not different from the medium-FS treatment in 2020 ($16.8 \pm 1.06$, $p = 0.38$) (Figure 4a) but was significantly less than the medium-FS treatment in 2021 ($16.1 \pm 1.15$, $p < 0.001$) (Figure 4b). The medium-FS treatment was significantly less than the high-FS treatment in 2020 ($20.2 \pm 1.45$, $p = 0.03$) but was similar in 2021 ($16.8 \pm 1.80$, $p = 0.78$). A study conducted in Spring Hill, TN, reported similar results where sunnhemp–corn, cowpea–corn, and crabgrass–corn intercrops were compared [22]. Although the sunnhemp alone produced higher biomass ($7.5$ Mg ha$^{-1}$) than cowpea ($3.4$ Mg ha$^{-1}$) or crabgrass ($2.8$ Mg ha$^{-1}$), the production of corn and total biomass accumulation were not affected by the inclusion of any of the forages. Compared to a study conducted by Bell et al., 2021 [35] in Bushland, Texas, DM production of sorghum at all seeding rates of sorghum and sunnhemp in the present study was within the range of DM produced by all varieties of monocrop sorghum ($13.7$ to $25.6$ Mg ha$^{-1}$).

(**a**)

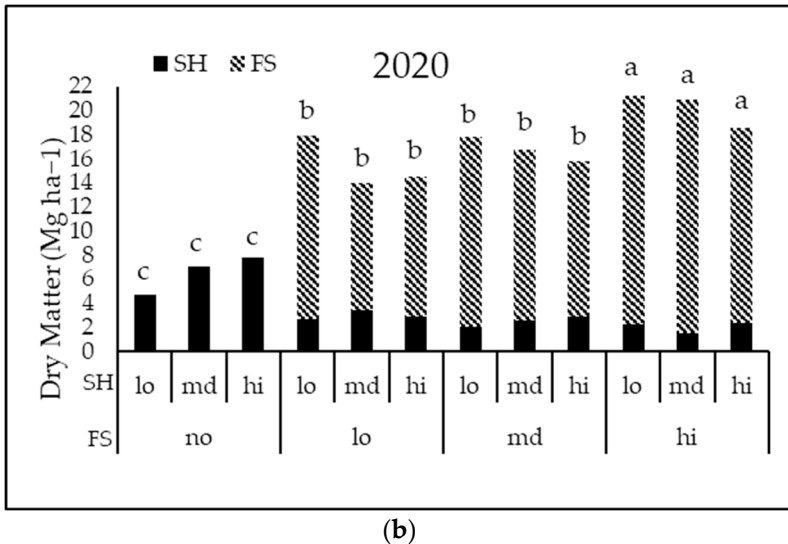

(**b**)

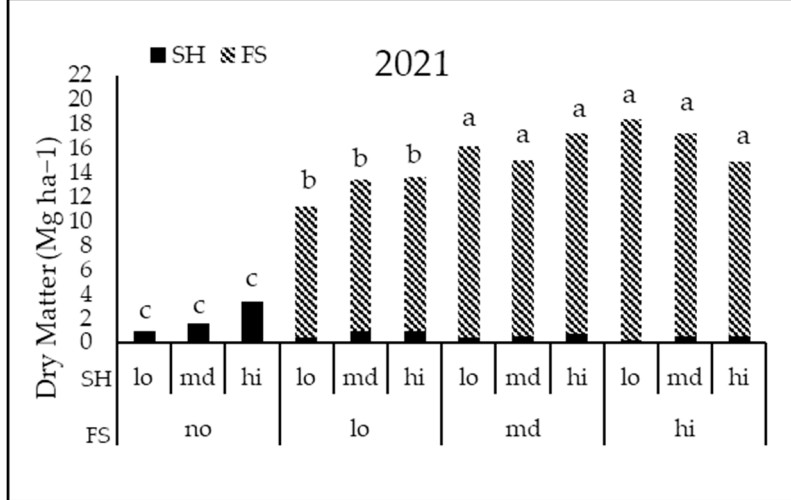

**Figure 4.** Dry matter accumulation (Mg ha$^{-1}$) of forage sorghum (FS) and sunnhemp (SH) in (**a**) 2020 and (**b**) 2021 when harvested 90 days after planting. Mean separation letters represent total DM in each treatment. Different letters within the same year are different at $p \leq 0.05$.

In 2020, the mean midseason harvest DM across all SH treatments was 3.6 Mg ha$^{-1}$ in the high forage sorghum seeding rate treatments, 2.6 and 2.7 Mg ha$^{-1}$ in the medium and low forage sorghum seeding rate treatments, respectively, and 1.1 Mg ha$^{-1}$ in treatments containing no forage sorghum (Figure 5a). In 2021, average DM across all SH treatments was different among all forage sorghum seeding rate treatments (*p*, 0.001). The highest DM was 4.4 Mg ha$^{-1}$ in the high forage sorghum seeding rate treatment (Figure 5b). The medium forage sorghum seeding rate produced 3.5 Mg ha$^{-1}$, the low forage sorghum seeding rate produced 2.6 Mg ha$^{-1}$, and the treatments with no forage sorghum produced 0.5 Mg ha$^{-1}$ DM. The lower DM production in 2021 can be explained by the slower accumulation of GDDs in 2021 than in 2020 (Table 1). In the sunnhemp monocrop (treatments with no forage sorghum), the average DM harvested at 45 DAP was 1.1 Mg ha$^{-1}$ which is the same as the DM amount harvested at 45 DAP in both years of research conducted by Lepcha et al., 2018 [29] in Columbia, MO, when plants were cut down to 10 or 15 cm above the soil surface.

(**a**)

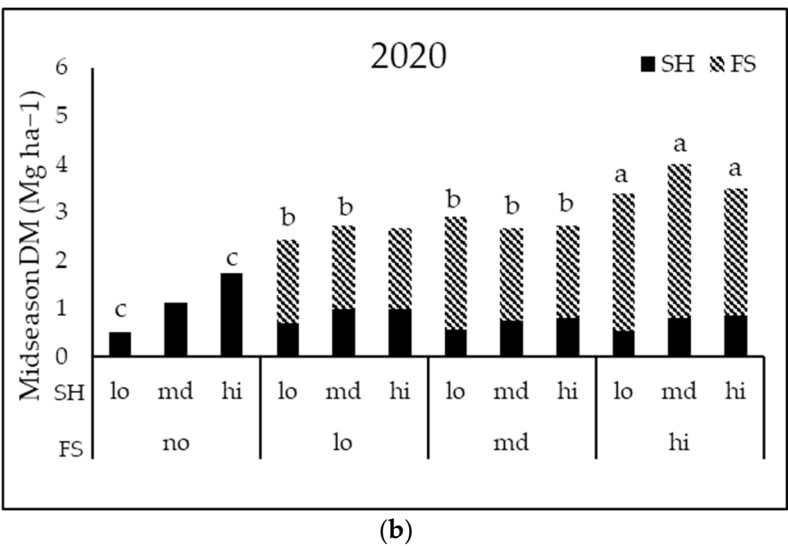

(**b**)

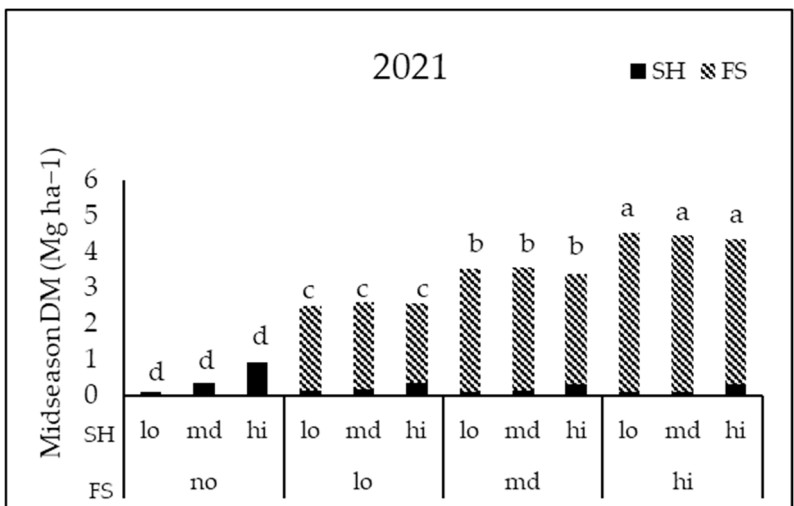

**Figure 5.** Midseason dry matter (DM) produced 45 days after planting in (**a**) 2020 and (**b**) 2021. Mean separation letters represent total DM in each treatment. Different letters within the same year are different at $p \leq 0.05$.

The average regrowth DM of sunnhemp was higher in 2020 than in 2021 due to the greater accumulation of GDDs in 2020 (Figure 6a,b). In 2020, the intercrop regrowth DM was 2.5 Mg ha$^{-1}$, and in 2021, the regrowth DM was 0.3 Mg ha$^{-1}$. Similar results were reported by Lepcha et al., 2018 [29] for the regrowth DM of sunnhemp with 0.8 Mg ha$^{-1}$ in 2014 and 0.2 Mg ha$^{-1}$ in 2015. Their results were also attributed to lower accumulation of GDDs in the second year.

There is potential for the forage sorghum/sunnhemp intercrop to be harvested midseason or possibly grazed down to 30 cm and still provide enough regrowth for an end-of-season harvest. The planting density of each species did not affect the forage quality or DM production in the regrowth harvest; however, higher total growing season DM was produced when the forage sorghum was seeded at the highest rate, 11.2 kg ha$^{-1}$.

(**a**)

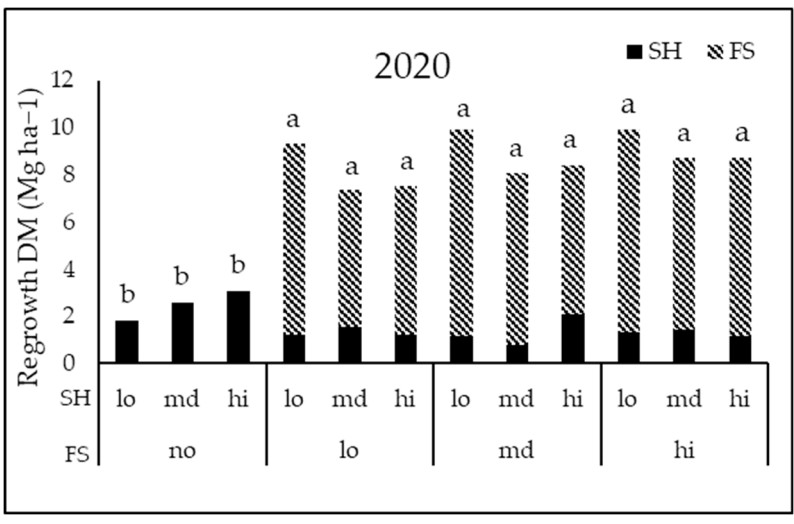

(**b**)

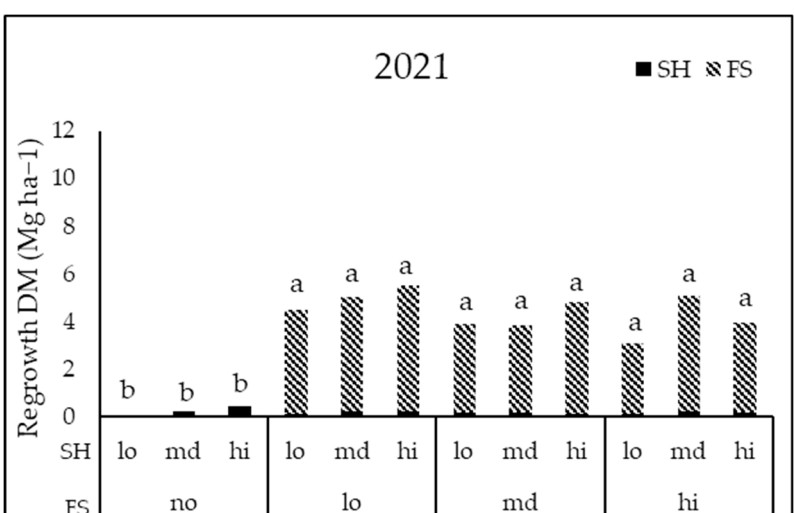

**Figure 6.** Regrowth DM produced in (**a**) 2020 and (**b**) 2021. Mean separation letters represent total DM in each treatment. Different letters within the same year are different at $p \leq 0.05$.

*3.5. Forage Nutritive Value*

There was no interaction of forage sorghum X sunnhemp seeding rate on forage quality. Additionally, there was no sunnhemp seeding rate effect; therefore, forage quality was analyzed by forage sorghum seeding rate only. Crude protein content was significantly higher in treatments containing no forage sorghum (8.1%) than in all other treatments in both years (Table 2). In 2020, CP was not different in low- and medium-forage-sorghum treatments with 5.1 and 5.0%, respectively. Compared to all other treatments, CP was lowest in the high-forage-sorghum treatment with 4.5%. In 2021, the only difference in CP occurred in the treatments with no forage sorghum, where 8.8% CP was reported compared to an average of 3.8% CP in all treatments containing forage sorghum. Another study conducted in the THP on forage sorghum–sunnhemp intercropping reported that CP increased when sunnhemp was grown alone, although the CP content found in their study was much greater [33]. Eberle and Shortnacy, 2021 [36] reported higher CP values in sunnhemp as well. Lower CP values in sunnhemp generally can be attributed to a low leaf-to-stem ratio [37]; however, this ratio was not calculated in this study.

**Table 2.** Total intercrop forage nutritive values for 2020 and 2021 by forage sorghum seeding rate near Canyon, TX, USA. Means for treatments containing different seeding rates of sunnhemp were nested in each forage sorghum seeding rate treatment due to no forage sorghum seeding rate X sunnhemp seeding rate interaction and no sunnhemp main effect on the response variables reported.

| FS Rate | CP [a] | ADF | NDF | TDN | RFV [b] |
|---|---|---|---|---|---|
| | --------------------------------%-------------------------------- | | | | |
| **2020 Growing Season** | | | | | |
| no | 8.1 ± 0.82 a | 55.8 ± 1.26 a (38.4) [c] | 69.4 ± 0.60 a | 40.2 ± 1.24 a (10.5) | 61.8 ± 1.39 a (12.8) |
| lo | 5.1 ± 0.47 b | 44.3 ± 0.22 b (19.0) | 67.5 ± 0.44 a | 53.0 ± 0.23 b (29.9) | 75.1 ± 0.29 b (28.1) |
| md | 5.0 ± 0.52 c | 44.3 ± 2.48 b (18.9) | 66.9 ± 1.42 a | 53.0 ± 2.65 b (29.7) | 75.8 ± 4.13 b (28.8) |
| hi | 4.5 ± 0.27 c | 44.5 ± 1.84 b (21.6) | 67.1 ± 2.25 a | 52.9 ± 2.15 b (28.0) | 75.3 ± 4.34 b (28.4) |
| *p* value | <0.001 | 0.001 | 0.706 | 0.001 | 0.010 |
| **2021 Growing Season** | | | | | |
| no | 8.8 ± 0.89 a (36.0) | 58.3 ± 0.14 a (36.0) | 67.8 ± 0.81 a (27.2) | 37.5 ± 0.26 a (6.0) | 60.3 ± 1.26 a (5.0) |
| lo | 4.0 ± 0.18 b (20.3) | 39.4 ± 1.33 b (16.2) | 60.1 ± 0.51 a (19.0) | 58.5 ± 1.46 b (26.3) | 89.9 ± 2.18 b (25.6) |
| md | 3.7 ± 0.62 b (14.8) | 38.5 ± 0.63 b (13.6) | 59.5 ± 0.26 a (14.6) | 59.4 ± 0.77 b (29.0) | 92.1 ± 1.01 b (30.9) |
| hi | 3.8 ± 0.38 b (14.8) | 40.4 ± 0.46 b (19.2) | 61.5 ± 0.65 a (24.6) | 52.4 ± 8.01 b (21.2) | 86.8 ± 0.95 b (21.0) |
| *p* value | <0.001 | <0.001 | 0.087 | <0.001 | <0.001 |

Note. Within columns, means followed by the same letter are not significantly different at the 0.05 probability level. [a] CP, crude protein; ADF, acid detergent fiber; NDF, neutral detergent fiber; TDN, total digestible nutrients; RFV, relative feed value; FS, forage sorghum. [b] Relative feed value measured using the equation RFV = (DDM × DMI)/1.29 as compared to alfalfa. [c] Numbers in parentheses display mean scores for data analyzed using chi-square.

Results obtained for ADF and NDF were similar between years (Table 2). Percent ADF was different between treatments with forage sorghum and without forage sorghum. The 2020 average was 44.4% ADF with forage sorghum and 55.8% without forage sorghum. In 2021, ADF was 39.4% with forage sorghum and 59.3% without forage sorghum. These values were similar to the ADF values reported in Eberle and Shortnacy, 2021 [36]. Higher ADF in the sunnhemp may be attributed to the maturity of the stems at harvest and a lower leaf-to-stem ratio, as suggested by the results reported by Mansoer et al. [37]. The exact stem-to-leaf ratio was not calculated in the current study, but it was noted that lower leaves in the sunnhemp had begun to senesce prior to harvest in both years, and grasshoppers defoliated some of the sunnhemp in 2021. Percent NDF did not differ among treatments in either year. The average NDF values were 67.7% in 2020 and 62.2% in 2021 which are too high to be considered fair-quality hay by USDA standards [38]. La Guardia Nave et al. [39] reported similar results when sunnhemp was intercropped with corn for silage, where the NDF of the sunnhemp–corn mixture was 61.5% when harvested in September 2016.

Total digestible nutrients were higher in treatments containing forage sorghum than treatments with no forage sorghum in 2020 and 2021 (Table 2). Relative feed value followed the same trend. However, no seeding ratio treatment in this study obtained an RFV ≥ 130, which is the lowest suggested value for fair-grade USDA hay quality for alfalfa and alfalfa mixes [38]. According to Mahfouz et al. [40], harvesting forage sorghum when it reaches boot stage rather than physiological maturity can improve CP content and overall forage quality due to the reduction in fiber.

Crude protein from the midseason harvest was highest in treatments with no forage sorghum in both years at 16.9 and 17.7% (Table 3). This is higher than the results reported

by Lepcha et al., 2018 [29] in the 45 DAP harvest with 10.1 and 14.3% CP in 2014 and 2015, respectively. In 2020, the treatments with forage sorghum were not different and contained an average of 12.1% CP. In 2021, CP was different between treatments with no forage sorghum and high forage sorghum. The low and medium forage sorghum seeding rates contained an average of 10.9% CP, and the high seeding rate was 9.5% (Table 3). Mahfouz et al. [40] found that percent CP can be increased in forage sorghum when harvested at boot stage rather than physiological maturity. In their study, CP averaged 3.2% higher when harvested at boot stage than when harvested at physiological maturity. This finding coupled with the midseason CP values for sunnhemp in this study suggests that rather than choosing harvest date based on the percent flowering of sunnhemp, a forage-sorghum–sunnhemp intercrop could yield higher CP when harvested at boot stage for forage sorghum.

**Table 3.** Midseason intercrop dry matter and forage quality for 2020 and 2021 by forage sorghum seeding rate near Canyon, TX, USA. No forage sorghum X sunnhemp interaction occurred and no difference was found between sunnhemp seeding rates, so response variables are reported by forage sorghum seeding rate only.

| FS Rate | CP [a] | ADF | NDF | TDN | RFV [b] |
|---|---|---|---|---|---|
| | --------------------------------%-------------------------------- | | | | |
| **2020 Growing Season** | | | | | |
| no | 16.9 ± 1.38 a (36.9) [c] | 37.9 ± 2.24 a | 43.3 ± 2.18 a (5.5) | 60.2 ± 2.74 a (20.8) | 128.3 ± 10.12 a (31.4) |
| low | 12.8 ± 1.19 b (21.3) | 39.1 ± 0.90 a | 59.1 ± 1.82 b (25.3) | 58.9 ± 1.00 a (22.0) | 92.3 ± 3.51 a (20.9) |
| med | 12.3 ± 0.83 b (18.5) | 39.2 ± 0.12 a | 59.4 ± 0.38 b (26.8) | 58.7 ± 0.12 a (21.4) | 91.3 ± 0.58 a (19.7) |
| high | 11.3 ± 0.50 b (12.6) | 38.6 ± 1.18 a | 60.4 ± 0.95 b (31.0) | 59.5 ± 1.56 a (23.8) | 91.3 ± 2.08 a (17.6) |
| *p* value | <0.001 | 0.771 | <0.001 | 0.958 | 0.077 |
| **2021 Growing Season** | | | | | |
| no | 17.7 ± 0.83 a (42.5) | 35.2 ± 0.50 a | 42.8 ± 1.04 a (6.5) | 63.2 ± 0.29 a (35.7) | 130.2 ± 6.00 a (42.5) |
| low | 11.3 ± 0.16 b (26.5) | 36.9 ± 0.54 b | 61.2 ± 0.25 b (24.1) | 61.2 ± 0.68 a (31.1) | 91.8 ± 0.90 b (25.6) |
| med | 10.4 ± 0.64 b (19.3) | 38.0 ± 0.80 bc | 62.0 ± 1.46 bc (29.0) | 60.0 ± 0.80 b (19.9) | 89.5 ± 2.70 b (20.7) |
| high | 9.5 ± 0.82 c (9.79) | 38.9 ± 0.52 c | 64.0 ± 1.60 c (38.4) | 58.9 ± 0.65 b (11.3) | 85.2 ± 2.92 c (9.2) |
| *p* value | <0.001 | <0.001 | <0.001 | <0.001 | <0.001 |

Note. Within columns, means followed by the same letter are not significantly different at the 0.05 probability level. [a] CP, crude protein; ADF, acid detergent fiber; NDF, neutral detergent fiber; TDN, total digestible nutrients; RFV, relative feed value; FS, forage sorghum. [b] Relative feed value measured using the equation RFV = (DDM × DMI)/1.29 to determine feed value as compared to feed value of alfalfa. [c] Numbers in parentheses display mean scores for data analyzed using chi-square.

In 2020, ADF was not different between treatments in the midseason harvest with an average of 38.7% across all treatments (Table 3). In 2021, ADF was lower in treatments without forage sorghum at 35.2% and highest in the high-forage-sorghum treatments at 38.9%. This is similar to results reported for forage sorghum by Atis et al. [41] where ADF was 35.5% when harvested at panicle emergence. Lepcha et al. [29] reported 38.0 and 44.2% ADF for sunnhemp harvested at 45 DAP which is higher than the 35.2% ADF average of sunnhemp treatments with no forage sorghum in 2021.

Midseason NDF was lower in treatments without forage sorghum than in treatments with forage sorghum in both years at 43.3 and 42.8%. In 2020, treatments containing forage sorghum were not different with an average of 59.6% (Table 3). In 2021, NDF in the

low-forage-sorghum treatments was different from the high-forage-sorghum treatments at 61.2% but not from the treatments containing medium forage sorghum. Medium- and high-forage-sorghum treatments were similar at 62.0 and 64.0% NDF, respectively.

Midseason TDN was not different in 2020 with an average of 59.3%. In 2021, TDN from the no- and low-forage-sorghum treatments averaged 62.2% and were different from the medium- and high-forage-sorghum treatments that averaged 59.5% (Table 3). Regrowth TDN was different among treatments with no forage sorghum and treatments with forage sorghum in both years (Table 4).

**Table 4.** Regrowth intercrop dry matter and forage quality for 2020 and 2021 by forage sorghum seeding rate near Canyon, TX, USA. No forage sorghum X sunnhemp interaction occurred and no difference was found between sunnhemp seeding rates, so response variables are reported by forage sorghum seeding rate only.

| FS Rate | CP [a] | ADF | NDF | TDN | RFV [b] |
|---------|--------|-----|-----|-----|---------|
| | --------------------------------%-------------------------------- | | | | |
| **2020 Growing Season** | | | | | |
| no | 8.9 ± 0.64 a (42.0) [c] | 56.8 ± 1.27 a (40.2) | 67.9 ± 0.40 a | 49.1 ± 1.22 a (8.9) | 61.9 ± 1.89 a (12.6) |
| low | 5.2 ± 0.45 b (18.5) | 46.3 ± 1.63 b (21.5) | 68.4 ± 1.97 a | 50.8 ± 1.84 b (27.7) | 72.3 ± 3.04 b (27.2) |
| med | 5.3 ± 0.10 b (18.5) | 44.5 ± 1.21 b (16.1) | 67.0 ± 0.45 a | 52.8 ± 1.36 b (32.9) | 75.3 ± 1.04 b (32.4) |
| high | 5.3 ± 0.25 b (19.0) | 45.8 ± 1.11 b (20.2) | 68.3 ± 1.29 a | 51.2 ± 1.29 b (28.6) | 72.7 ± 1.23 b (25.8) |
| *p* value | <0.001 | <0.001 | 0.774 | <0.001 | 0.005 |
| **2021 Growing Season** | | | | | |
| no | 10.1 ± 0.20 a (31.5) | 53.7 ± 2.40 a | 66.2 ± 0.87 a (14.8) | 42.8 ± 2.71 a (4.5) | 66.5 ± 2.18 a (4.5) |
| low | 5.1 ± 0.18 b (13.8) | 40.7 ± 0.61 b | 63.3 ± 0.83 a (22.5) | 57.1 ± 0.79 b (26.2) | 84.2 ± 0.54 b (24.3) |
| med | 5.5 ± 0.48 b (16.4) | 42.2 ± 0.94 b | 63.3 ± 1.58 a (23.7) | 55.4 ± 1.18 b (20.8) | 82.4 ± 3.24 b (20.9) |
| high | 5.1 ± 0.31 b (15.1) | 41.9 ± 0.60 b | 62.6 ± 1.00 a (19.9) | 55.6 ± 0.57 b (21.7) | 83.5 ± 1.98 b (23.4) |
| *p* value | 0.014 | 0.008 | 0.331 | <0.001 | <0.001 |

Note. Within columns, means followed by the same letter are not significantly different at the 0.05 probability level. [a] CP, crude protein; ADF, acid detergent fiber; NDF, neutral detergent fiber; TDN, total digestible nutrients; RFV, relative feed value; FS, forage sorghum. [b] Relative feed value measured using the equation RFV = (DDM × DMI)/1.29 to determine feed value as compared to feed value of alfalfa. [c] Numbers in parentheses display mean scores for data analyzed using chi-square.

Midseason RFV was not different between treatments in 2020. In 2021, RFV trended downward with added amounts of forage sorghum (Table 3). The RFV was highest in treatments with no forage sorghum at 130.2 and lowest in the treatments with high forage sorghum at 85.2. Treatments with low and medium forage sorghum averaged 91.8 and 89.5, respectively. These results are similar to the results reported by Atis et al. [41] where average RFV was 85.2 when sorghum was harvested at panicle emergence and Machicek et al. [11] where forage sorghum harvested 45 DAP had an average RFV of 86.6.

Percent CP decreased from the 45 DAP to the 90 DAP harvest of regrowth in all treatments (Table 4). Treatments with no forage sorghum maintained the highest CP. The only difference between treatments for 90 DAP CP in the no-forage-sorghum seeding rate treatments at 8.9 and 10.1% in 2020 and 2021, respectively. Similarly, Lepcha et al. [10] reported 9.6 and 12.4% CP on the regrowth of sunnhemp, which decreased from the initial harvest 45 DAP. One probable cause for this was the decrease in leaf/stem ratio since CP is typically more highly concentrated in the leaves of sunnhemp plants [12]. The average CP of the regrowth from treatments containing forage sorghum was 5.3% in both years

(Table 4). This is lower than the 6.6% CP found in the ratoon crop of forage sorghum from Machicek et al. [11] when harvested at 90 DAP, after cutting 45 DAP. This is also lower than the percent CP reported by Contreras-Govea et al. [4] when sorghum was intercropped with different legumes and harvested at either soft-dough or black-layer stages of sorghum. In the soft-dough harvest, CP averaged 9.9% from sorghum as a monoculture and 10.5% from sorghum mixed with each of the legumes. In the black-layer harvest, the sorghum monoculture contained 7.5% CP, and the intercrops ranged from 6.9 to 9.4% CP. The lower percent CP at the later harvest in their study is similar to the sorghum–sunnhemp intercrop.

The average ADF for treatments without forage sorghum was 56.8% in 2020 and 53.7% in 2021 (Table 4). The ADF for treatments containing forage sorghum was lower in both years with an average of 45.5% in 2020 and 41.6% in 2021. For the sunnhemp treatments containing no forage sorghum, ADF results from 2020 were higher than the ADF reported by Lepcha et al. [10] in both years at 46.1 and 53.2%. In 2021, the 53.7% ADF was similar to the second-year ADF (53.2%) reported by Lepcha et al. [10]. Treatments with forage sorghum in this study had higher ADF values than reported by Contreras-Govea et al. [4] when forage sorghum was intercropped with warm-season annual legumes which averaged 31.3%.

The NDF of plants in the regrowth harvest was not different between treatments in either year. The NDF average was 67.9% in 2020 and 63.9% in 2021 (Table 4). This is slightly higher than the results reported by Machicek et al. [11] where the regrowth of forage sorghum averaged 62.1% in 2016 and 57.2% in 2017. Results obtained from Lepcha et al. [10] for sunnhemp regrowth were lower at 57.7% in 2014 and similar at 64.2% in 2015.

In both years, regrowth RFV was different in treatments without forage sorghum than treatments with forage sorghum. Treatments with no forage sorghum averaged 61.9 in 2020 and 66.5 in 2021, and treatments with forage sorghum averaged 73.4 in 2020 and 83.4 in 2021 (Table 4). This is lower than the RFV reported in the study by Machicek et al. [11] where the regrowth of sorghum averaged 86.5 and 96.3 RFV.

## 4. Conclusions

Growing sunnhemp as a monocrop or intercropped with forage sorghum was a viable forage option in the Texas High Plains, especially when adequate seasonal GDD accumulation occurred during vegetative growth. Based on seasonal average temperatures in the region, adequate GDD accumulation should be easily obtained in most years. However, sunnhemp as a monocrop produced significantly less DM than when intercropped with forage sorghum. Although no difference in dry matter production was observed among seeding rate treatments of sunnhemp with forage sorghum, a seeding rate of 33.6 kg seed ha$^{-1}$ sunnhemp would be recommended to provide enough cover to maximize light interception and contribute to the forage quality of the intercrop.

Increasing row spacing between forage sorghum and allowing more rows of sunnhemp at a greater seeding rate in between rows of forage sorghum might improve forage quality without reducing DM production. Additional studies could evaluate the grazing potential of the intercrop by incorporating livestock onto the field when plants have reached 60–90 cm or 45 DAP in order to maximize sunnhemp stem:leaf ratio. In years when GDD accumulation is not sufficient for grazing, harvesting the intercrop when the sorghum reaches boot stage has the potential to increase CP content and overall RFV by reducing fiber content in the stems and decreasing the stem–leaf ratio.

**Author Contributions:** Conceptualization, H.M.M. and S.A.O.; Methodology, S.A.O., B.C.B. and H.M.M.; Formal Analysis, H.M.M. and M.B.R.; Investigation, H.M.M.; Writing—Original Draft, H.M.M.; Writing—Review and Editing, H.M.M., S.A.O., B.C.B. and M.B.R.; Funding Acquisition, S.A.O. All authors have read and agreed to the published version of the manuscript.

**Funding:** Funding in part was from the Ogallala Aquifer Program, a consortium between USDA-Agricultural Research Service, Kansas State University, Texas AgriLife Research, Texas AgriLife

Extension Service, Texas Tech University, and West Texas A&M University. Partial funding was paid by the Vernon Harman Professorship at West Texas A&M University.

**Data Availability Statement:** The data presented in this study are available on request from the corresponding author. The data are not publicly available as the data produced are the property of West Texas AA&M Univiersity.

**Conflicts of Interest:** The authors declare no conflict of interest.

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
