# Peer review of "Intercropping Forage Sorghum with Sunnhemp at Different Seeding Rates to Improve Forage Production"

_agronomy, doi:10.3390/agronomy13123048_

Round 1

Reviewer 1 Report

Comments and Suggestions for Authors

Dear Authors,

The subject matter of the field research undertaken in my opinion is very interesting which can contribute to the spread of intercropping which guarantees good quality fodder and is more environmental compared to cereal crops. In the results and discussion section, references to the results of other researchers and the supposed mechanisms of occurrence are missing in several places. Some of these are indicated, but I suggest an analysis of the entire section and similar additions to all paragraphs. I have included a detailed reference to the content of the manuscript below.

L22 It would be better if the keywords did not repeat the words from the title

L24 In my opinion, a good addition to the manuscript would have been a description of intercropping due to plant selection and component share of seeding. The authors test the amount of seed sowing I suspect because of the min. possible influence of competition and the mentioned yield of good quality forage. Intercropping with legumes also has a number of other benefits that it would be good to mention min. reducing the use of mineral fertilizers, less weed pressure and maintaining yield stability under different conditions. Also, due to the fact that irrigation is being tested, it would be good to state how the plants tested behave under water deficit conditions to emphasize the point of irrigation.

L73 should be given in terms of N kg/ha

L83-84 if possible should be given CFU/g

L84-88 In my opinion, the amount of seed sown per ha should also be reported. Legumes in particular are characterized by a variable weight of 1000 seeds from year to year so with the same seeding rate of kg from year to year, the actual plant density could be variable. In addition, the selection of components for intercropping is also puzzling to me: 1. why sorghum was not sown without sunnhemp as a reference to improve the quality of the forage obtained? 2. with the analyzed seeding combinations there were plots where, for example, 11.2 kg of sorghum and 50.4 kg of sunnhemp were sown and, on the other hand, 2 kg of sorghum and 16.8 kg of sunnhemp so significantly different plant density per hectare. So a smaller number of plants will generally indicate a smaller DM. Of course, there is the competitiveness of the crop, possible better growth with lower planting density and possible better quality with less nutrient availability. Nevertheless, I would like the authors to explain this approach to crop seeding. 3. lower plant density per hectare certainly resulted in less soil cover thus a greater possibility of soil drying during the season and the possibility of greater weed growth due to less competition of crops with weeds for resources (nutrients, light, water). The authors make no mention of weed control in crops. Was there a weed problem? A similar question applies to the sowing of sunnhemp only since legumes generally have an intense weed problem.

L157 Why was there no additional analysis of the effects of growing season conditions on crops and interactions with other factors, and the timing of crop harvesting and interactions with other factors?

L181 There is no reference to results obtained by other researchers in similar intercropped crops. The authors mention the lack of field studies of this mixture but I believe that it can be compared for intercropping with other similar legumes. This would show whether sunnhemp responds better or worse to intercropping

L225 As before, there are no references to results from other researchers. In addition, there is also a lack of putative mechanisms for such results. I suspect that simply a higher plant density shows a much higher demand. Additionally, one would have to address the differences between sorghum and sunnhemp crops

L317-318 A few lines earlier it was found that the amount of sunnhemp seeding had no significant effect on forage quality

L321 And how did this compare to the results of other researchers?

L345 In addition, one could include the results obtained in harvesting at an earlier stage which also suggest that harvesting at the stage presented in this paragraph was relatively late due to the higher NDF and ADF in sunnhemp compared to sorghum. Reference can also be made here to studies by other authors showing the effect of developmental stage on NDF ADF RFV values

L366-370 I agree with this statement, however, when analyzing this together with DM yields, it is not beneficial because the amount of DM is much lower at this stage. So such a treatment would not be economically justified.

L473 In the results obtained, there was no effect of sunnhemp seeding on forage quality.

Author Response

The comments that are not directly addressed below were included in the highlighted revisions of the manuscript.

L24 …due to the fact that irrigation is being tested, it would be good to state how the plants tested behave under water deficit conditions to emphasize the point of irrigation.

Some information on irrigation was added to the introduction, however, I wanted to point out that nearly all crops in the Texas High Plains are irrigated due to high temperatures, low humidity, and lack of seasonal precipitation in the area. We didn’t test different levels of irrigation in this study, although this would have been an interesting variable to include. I hope to conduct future studies in the area that an address more questions involving this cropping system.

L157 Why was there no additional analysis of the effects of growing season conditions on crops and interactions with other factors, and the timing of crop harvesting and interactions with other factors?

Irrigation was evenly applied to limit any potential water factors. Timing of harvest was not compared because we felt that the harvest times were not implemented to compare across timing, but rather to determine whether regrowth can occur to provide multiple cuts of forage for livestock in the area. Due to the limited space allotted for this experiment, plot size was limited to be able to include all treatments that were evaluated.

Reviewer 2 Report

Comments and Suggestions for Authors

General comment

The manuscript is an original study on forage sorghum and sunnhemp intercropping to improve forage production.

The topic is interesting but some paper sections such as introduction, results and discussion should be improved. The results highlight that forage sorghum is very competitive and do not allow a proper growth of the intercropped sunnhemp. In my opinion, in the second year of the experiment this should have been fixed by introducing an additional treatment concerning a lower sorghum seeding rate.

The introduction section is too poor. This section should be a strong justification for your study. You should give a strong explanation on why intercropping is important, the benefit and the flaws by extensively citing references on other type of intercropping. Please follow some more detailed information belowin the specific comments.

In discussion section you should discuss your results and compare them with references. Please improve it. I think adding land equivalent ratio to the manuscript will improve it as far as is one of the most fitting parameters used to compare sole crop with intercropping systems.

I propose major revision

Specific comments

Page 1 Line 32: It’s better to indicate the average annual depletion rate.

Page 1 Line 37: Also the different rooting system can be more efficient in using water from different soil layers

Page 1 Line 42: what about the content in alkaloids? mention that some varieties have these problem

Page 2 Line 47: Implement this section by introducing also forage sorghum characteristics and highlighting the benefit from intercropping more in detail. There is a published study on the same system (sunn hemp and biomass sorghum): Zegada-lizarazu, W., Parenti, A., & Monti, A. (2021). Intercropping grasses and legumes can contribute to the development of advanced biofuels. Biomass and Bioenergy, 149, 106086. https://doi.org/10.1016/j.biombioe.2021.106086

Page 2 Line 73: 196 units of Nitrogen or Urea?

Page 2 Line 80: that seems a bit low. Then if you split in half you harvested only 1.5 m2. What about border effect?

Page 6 Line 216: In 2020 the sunnhemp sorghum IWUE does not increase with seeding rate. No differences were found between md and hi sorghum densities. Did you look for correlation between IWUE and seeding rate?

Page 7 Figure 2: Letter a is usually for the highest value, please correct

Page 7 Line 241: Balkcom study is on sunnhemp as cover crop. Improve discussion by comparing with similar system such as the one presented in: Zegada-lizarazu, W., Parenti, A., & Monti, A. (2021). Intercropping grasses and legumes can contribute to the development of advanced biofuels. Biomass and Bioenergy, 149, 106086. https://doi.org/10.1016/j.biombioe.2021.106086

Page 9 Line 292: I'm not sure that 'regrowth DM' is correct

Page 11 Table 2: do not repeat DM results in the table if they have already been presented above

Page 15 Line 486: I don't see grazing as feasible. The problems are: i) trampling; ii) what about monocrotaline in young sorghum leaves?

An additional risk is that livestock would prefer the soft sunnhemp leaves in spite of sorghum, which can lead to an even more unbalanced intercropping

Author Response

The comments that are not directly addressed below were included in the highlighted revisions of the manuscript.

Page 1 Line 42: what about the content in alkaloids? mention that some varieties have these problem

Sunnhemp does not generally pose an issue with alkaloid content like other species within the Crotalaria genus, therefore was not addressed in this text.

Page 15 Line 486: I don't see grazing as feasible. The problems are: i) trampling; ii) what about monocrotaline in young sorghum leaves?

I did end up removing the suggestion for grazing potential from the conclusion section, but I do think this could be a future experiment on this cropping system in the area. Most of the crops grown in the Texas High Plains are for livestock feed, specifically beef and dairy cattle, so alternative forage options remain a priority for researchers and producers in the area. Trampling may be a concern when grazing, however, grazing mixed crops and other grasses in the area is a common practice. Monocrotaline does not exist in sorghum leaves. However, it may be found in sunnhemp leaves since it is a species in the genus Crotalaria, but sunnhemp doesn't produce enough alkaloids to pose any concern for livestock toxicity.

Reviewer 3 Report

Comments and Suggestions for Authors

Obtaining the right amount of good quality feed is an important element of animal production. A growing problem in many regions of the world affecting feed production are rainfall shortages and droughts. Therefore, possibilities are being sought to alleviate these problems, e.g. by introducing new plants into cultivation that are more resistant to water or heat stress. One such plant is sunnhemp. Intercropping is used in many countries. It has many advantages and is one of the elements of sustainable or organic agriculture, as it allows for more efficient use of environmental resources. The effectiveness of intercropping depends on many factors, one of which is the proper selection of crop species and their distribution, including the density and the spacing between plant rows. Therefore, I think that the problem raised in the research is current and interesting. The main issue is to assess the possibility of intercropping of forage sorghum and sunnhemp in semiarid climate conditions and the impact of this cultivation on yield and feed quality depending on the harvest date

The issue presented in the manuscript is original and important. Manuscript will be an interesting source of information for both scientists and farmers dealing with forage production in semiarid condition.

Generally, the research presented in the manuscript was well planned and performed. However, in my opinion, it is a mistake not to introduce the sorghum facility in sole sowing. This would allow us to check how cultivation with sunnhemp affects the development and yield of this plant. Will the introduction of a plant from the legume family, which has the ability to assimilate atmospheric nitrogen, have a positive effect on the yield and its quality? It is known that sorghum has a higher yield potential than sunnhemp and in intercropping sorghum will be the basic plant. Therefore, such a variant with the introduction of sorghum without sunnhemp would be much more interesting.

Manuscript is generally well prepared, although after reading it, many comments and questions arise

Detailed comments:

1. Line [19-20] is ‘Results indicated that a sunnhemp-forage sorghum intercrop produced dry matter comparable to forage sorghum when sufficient heat units were obtained in the growing season’….  This is not clear from the research presented in the manuscript. There was no sorghum cultivation without sunnhemp.

2. The 'Introduction' chapter should be improved. It is short and quite general. More information should be provided about mixed cultivation of sorghum with other plants and the impact of various factors, including sowing rate and harvest date, on yield and sunnhemp quality. The authors used only 18 literature items in the manuscript, including 4 general ones, and yet in the scientific literature you can find many more studies regarding both mixed cultivation of sorghum and sunnhemp.

3. The 'Material and methods' chapter requires completion:

-          Please provide and characterize soil properties, granulometric composition, pH, organic matter content, macroelement content, etc

-          In line [72-73] a dose of nitrogen was given, which was completely applied before sowing. Is this the standard procedure used in sorghum cultivation? What about fertilizing with other ingredients, i.e. phosphorus, potassium? Wasn't used?

-          Is such a large dose of nitrogen appropriate for a legume such as sunnhemp?

-          please specify what the forecrop was

-          line [80] the plot area was small. Only 3.05 m2, why? Additionally, it was divided in half for different harvest dates, which means that there was only 1.53 m2. And in lines 128-129 the authors write that ‘..Two biomass samples from two, 1 m2 quadrats of each half-plot were collected…’. How, since it was only 1.53 m2. Same note about the line[134-135].

-          Line [81-82] ‘..North or south half of the plot was randomly chosen to hand….’ - how was one of the two randomly chosen?

-          Line [86] - please specify how many rows of sorghum and how many rows of sunnhemp were in one plot.

-          Line [88] - proposes to use different designations for sorghum and sunnhemp sowing rates. Using the same words - lo, md, hi - can be confusing for the reader. Please also explain on what basis such sowing standards for sorghum and sunnhemp were established.

-          Line [89] - how to sunnhemp hay on plots where sorghum was not sown? how many rows at what distances

-          Line [97-101] - What about weeds? Was there no need to limit them?

-          Line [125] - Please indicate what stage of development the sorghum and sunnhemp were in at the time of harvest 45 DAP. This is important information for other researchers working on this topic

-          Line [128-129] - it must be stated that after harvesting the sorghum and sunnhemp plants were separated and the dry weight was determined separately for these species. Please also explain how the area of 1 m2 was determined. what were its dimensions? Did it include two rows of sorghum and three rows of sunnhemp in between?

-          Line [131] – CP? - add ‘crud protein’

-          Line [145] the equation for calculating TDN includes FA (lipid content or crude fat content), and the methodology does not provide information that it was determined and by what method. How was NDFDp determined/calculated? I did not find such information below.

4. Results and Discussion

- This chapter needs to be improved. In subsection 3.1., 3.2., 3.3. there is no discussion of the results obtained at all. In the remaining subsections, there is often only a reference that someone else had a similar experience, without any attempt to explain the observed changes

- There are also interpretations of the results that raise doubts, e.g. Line [361-362] is ‘In 2021, CP was different between treatments with no forage sorghum and high forage sorghum’ - As the data in Table 3 show, the differences between no forage sorgum and low and medium forage sorgum were also significant.

      5. The ‘Conclusion’is too extensive. It should be shortened. Some information should be moved to the 'Results and Discussion' section, for example: line [473-477].

Author Response

The comments that are not directly addressed below were included in the highlighted revisions of the manuscript.

  1. Line [19-20] is ‘Results indicated that a sunnhemp-forage sorghum intercrop produced dry matter comparable to forage sorghum when sufficient heat units were obtained in the growing season’….  This is not clear from the research presented in the manuscript. There was no sorghum cultivation without sunnhemp. Many studies have been conducted on sorghum growth in the area, therefore a monocrop sorghum treatment was not included in this study due to limited space of the field plot. However, to address this comment, data from other studies have been inserted in the text to justify this claim.
  2. The 'Material and methods' chapter requires completion:

-          Please provide and characterize soil properties, granulometric composition, pH, organic matter content, macroelement content, etc A series description of the soil was provided which one can use to search for the specific characteristics outlined in this comment.

-          In line [72-73] a dose of nitrogen was given, which was completely applied before sowing. Is this the standard procedure used in sorghum cultivation? What about fertilizing with other ingredients, i.e. phosphorus, potassium? Wasn't used? Only N fertilization was required based on recommendations for sorghum growth which is common for sorghum cultivation in the Texas High Plains.

-          Is such a large dose of nitrogen appropriate for a legume such as sunnhemp? To avoid any variation in results of sorghum and sunnhemp growth at different seeding rates, we opted to supply the entire field with the amount of N that was recommended for sorghum growth.

-          Line [81-82] ‘..North or south half of the plot was randomly chosen to hand….’ - how was one of the two randomly chosen? Random choosing of plot half to cut was determined by flipping a coin, where heads indicated N, and tails indicated S. We determined that the simplicity of this method did not need inclusion in the text.

-          Line [88] - proposes to use different designations for sorghum and sunnhemp sowing rates. Using the same words - lo, md, hi - can be confusing for the reader. Please also explain on what basis such sowing standards for sorghum and sunnhemp were established. We chose to use these abbreviations for each seeding rate due to lack of space in charts, but each are clearly outlined in the charts to show which “lo, md, hi” belong with which species.

-          Line [145] the equation for calculating TDN includes FA (lipid content or crude fat content), and the methodology does not provide information that it was determined and by what method. How was NDFDp determined/calculated? I did not find such information below.

These values were determined in the laboratory analysis, not done by the specific researchers in this paper. The calculations used were provided, however, variables such as FA and NDFD were not provided.

Reviewer 4 Report

Comments and Suggestions for Authors

The paper “intercropping forage sorghum with sunnhemp at different seeding rates to improve forage production” presents data on the effect of sunn hemp and forage sorghum seeding rates on water use efficiency, light interception, biomass production, and forage quality of 45 day, 90 day, and 45-90 day regrowth. Generally the manuscript includes  valuable data that will be of interest to the scientific community, however there are some significant edits that need to be made prior to publication.

First, ANOVA tables need to be presented for the analyses that were run. These are critical for readers to be able to assess and interpret the results.

Second, I think the presentation of the data can be simplified to make the findings more clear. Data presented in figures 3, 4, and 5 is also presented in Tables 2, 3, and 4. The only additional data shown in the graphs are means for each SH SR within the different FS SR. The effect of SH SR is stated as being NS as is the interaction term so this data does not need to be presented. The other data shown in the graphs is the ratio of SH:FS biomass which was not analyzed and is not presented in the results. If the authors feel presenting this data is of importance then it should be analyzed and included in the results and discussion.

Third, the post hoc analysis should only be applied on the significant effects as was done in tables 2-4. If you leave the tables as is and delete the figures the post hoc analysis is correct but if you keep the figures to present the ratio of biomass between the two crops then the stats need to be re-run on the correct dataset.

Additional comments related to specific items in the text are listed below.

Introduction

Line 32 are their more current values than those from 2015?

Lines 35-37 the references used here are for forage sorghum with forage legumes, not sunn hemp. Remove the “such as sunn hemp” as it indicates these references report those results.

Line 45 In the US, Washington, Oregon, Idaho, Montana, Wyoming, Nevada, Utah, Colorado, Arizona, New Mexico, and Texas all have regions with a semi-arid climate. There are publications out of TX and WY on sunn hemp, both in semiarid locations. The authors should update their references to be sure relevant publications from these semiarid locations are included as it would add to the intro and discussion, at the very least they need to correct their statement as it is not accurate.

Materials and Methods

Line 73 was the same amount of urea applied to the monocrop sunn hemp plots and across all seeding rates? How would this affect biomass and protein at the higher seeding rates of FS? Were the lower rates overfertilized or were the higher rates under fertilized?

Lines 78-82 wording of the experimental design could be clarified. There were 12 treatments and 4 replicates arranged in a RCBD? then the split plot was a 45 and 90 day harvest in each main plot?

Line 80: Authors should provide the plot dimensions instead of the area. 3m2 would indicate plots were 1m x 3m (or something to that effect)? Or did the authors mean the plots were 3.05 m x 3.05 m (9.3m2)?

Line 83 need the rate of inoculum per mass of seed.

Lines 128-129 two biomass samples from two, 1m2 quadrats of each half-plot were collected… this means that you harvested 2m2 at 45 days and 2m2 at 90 days but your plots were only 3.05m2. Something isn’t adding up. If the plots were actually 3.05m x 3.05m then this statement makes sense.  

Lines 130 and 135 It is not clear how the biomass was handled. The data in the results includes sunn hemp biomass and forage sorghum biomass for each treatment but the methods don’t indicate that the biomass was ever separated post harvest. It also appears all the analysis is for the total biomass and not for the two fractions within in treatment.

Lines 158-165 Was year tested to determine if it was a significant variable before running the two years independently? If year was not significant in the model then you would just be able to report the combined data over two years. If this was evaluated it should be stated. If it wasn’t evaluated it should be. How was the split plot design accounted for in the analysis? Was the ‘nested’ arrangement accounted for as a random variable?

Results and Discussion

General comment: ANOVA tables for the main effects and interactions need to be presented for all data.

For light interception were any stats run on the data to test if FS or SH seeding rate effects or the interaction were significant? How was it determined to only show the FS seeding rate data in the graphs? This needs to be clarified in the results.

Line 182 and 187 remove “linearly”, with only two data points and no regression analysis it can not be stated that the increase over this time period was linear, just that there was an increase in PAR.

Lines 189-192 what is the explanation for this result, this was not seen in 2020?

Line 194 without a FS with no SH treatment its hard to know the contribution of the FS in light interception. In 2020 no FS had the same trends as the 3 treatments with FS, maybe slightly lower but without stats hard to know if its significant but its clear the SH contributes significant light interception. Even in 2021 the SH treatment would be contributing to half the light interception in the treatments with FS. There are likely other studies you can reference on FS light interception and if you would expect the monocrop interception to be as high as the values with the SH intercrop. This would be informative to add to the discussion.

Lines 211-213 the value of the error bar needs to be stated, are they standard error, standard deviation, or something else?

Figure 2. since the SH seeding rate is NS only the means for each FS treatment should be presented and the post-hoc analysis should only be done on those means (4 total) not across all treatments (12), this applies to figures 3 and 4 as well.

Line 222 since the SH no FS treatment produced significantly less biomass what do you think the chance is that it was over watered and the IWUE is misrepresenting the efficiency of the treatment since irrigation was not based on the needs of the individual plots? This could be part of your discussion and I am sure you could find references to help explain the data.

Section 3.4 it appears that the analysis was only done on total dry matter not the SH and FS fractions of each treatment. Without analyzing these you can not present results on them but instead need to present results on total dry matter which is what you analyzed. The dry matter results should be presented in the same way that the forage analysis is presented, as the total sample. Unless the authors re-run the stats to support the current presentation. Figures also need to be updated to reflect the data findings being reported. (see overall comments above)

Lines 236-237 this data is not significant and should not be presented as a significant finding.

Lines 237-238 There is no data presented on SH dry matter stats for treatments with FS, without a separate analysis for this the statement can not be supported.  

Lines 292-294 There is no data presented on SH dry matter stats for treatments with FS, without a separate analysis for this the statement can not be supported.

Section 3.5 The tables do not need to include ± standard error or standard deviation (its not stated what the value is), the post hoc category is sufficient. Removing these values will simplify the tables and make them more easily interpretable. If the value is left in it needs to be stated in the table legend what it is.

Line 311-356 It needs to be explicitly stated that this data is for the full season (90 day) harvest. It is not clear in the text or the table.  

Line 312 the authors need to discuss the performance of the SH only treatment to values reported in other studies. At a DM of 6.5 kg/ha is a CP of 8 in the range of what would be expected for SH? This comment applies to the ADF, NDF, TDN, and RFV as well. The readers want to know if SH in this growing region has the same quality as what is reported in other studies and growing areas.

Line 410 CP decreased compared to what?

I feel a discussion related to the trade off of DM production and forage quality would add relevance to the paper. A 45 day harvest and regrowth harvest will produce less DM than the 90 day harvest but the forage quality was higher, is it worth the trade off? What management system would be preferable?

Conclusions

Line 470-473 the recommendation here is not supported by the data. The light interception was not presented for the SH seeding rates so there is nothing to indicate a better cover. Additionally, SH seeding rate was NS for all variables including biomass and feed quality so why would a grower spend more money on seed?

Line 473-476 you have data on light interception of SH x FG SR correct? this is why you need to present the anova tables. If SH SR was NS then this should not be a major conclusion of your paper. You could possibly move your observations to the results and discussion section, but you need to draw your conclusions on the supportable results you present.

Line 476 I suggest taking out the reference to the cost of sunn hemp seed, there is no cost comparison per unit production presented and its just not necessary. Stick to conclusions related to data.

Lines 480-482 rephrase for clarification. Beginning of the statement refers to regrowth harvest but the end o f the statement says DM was higher in 11.2 kg/ha FS seeding rate but that was not the case for the regrowth so what data is being referenced?

Author Response

The comments that are not directly addressed below were included in the highlighted revisions of the manuscript.

General comment: ANOVA tables for the main effects and interactions need to be presented for all data. ANOVA tables are not often reported in agronomic studies, and given the detail in many other comments, there was not time to create these tables in a presentable format within the timeframe I was given to improve this manuscript.

Line 222 since the SH no FS treatment produced significantly less biomass what do you think the chance is that it was over watered and the IWUE is misrepresenting the efficiency of the treatment since irrigation was not based on the needs of the individual plots? This could be part of your discussion and I am sure you could find references to help explain the data. Some data from a subsequent study has been added to the discussion of IWUE that reflects the findings in this study. Although I do feel that more experiments should be conducted in the area to support these claims and further flesh out any changes in IWUE on this cropping system.

Reviewer 5 Report

Comments and Suggestions for Authors

Specific comments

As a researcher in the field of farming, I am very interested in your work. I have looked thoroughly at your article and I see that you did a lot of work on it.

However, There are some problems in the article that need to be solved, if I understand your description correctly. As far as I see, the paper can be accepted if the points below are dealt with appropriately

Abstract

1. Line 8. It is suggested to verify the correct writing of scientific names.

2. It is recommended to put different treatments after the background.

3. Try to select keywords different from the title of the article for retrieval.

4. Line 13, “semiarid” should change  semi-arid”.

5. In the abstract, it is suggested to simplify the background, which can be shown in the introduction section.

6. Line 12-15, this sentence is suggested to be deleted and added in the introduction.

7. Suggest adding some data to the result.

8. Add the test place and the specific year of the experiment to the abstract.

9. The analysis of results in the abstract is different from the purpose of this study and lacks the analysis of results for specific values.

10. The abstract lacks a description of the conclusions and does not indicate which of the seeding ratios worked better.

Keywords

11. Please re-describe the keywords, "forages" and "sorghum" should be one keyword, it is recommended not to write them separately.

Introduction

12. Line 25, “semiarid” should change  semi-arid”.

13. The introduction is too little and not very sufficient.

14. It is suggested to add specific research direction and content.

15. Line 26-27,(Sorghum bicolor [L.] Moench)or [Sorghum bicolor (L.) Moench]please unify the format.

16. Line 35-36,sunnhemp (Crotolaria juncea L.)is different from sunnhemp (Crotalaria juncea L.)in abstract line 10.Please check carefully whether the Latin name of the plant is correct.

17. Line 39, why was "Sunnhemp" selected and has it been grown for many years in the test area?

18. Line 48, suggests a hypothesis for the relevant results, which can be tested.

19. Line 48. Based on previous studies, put forward reasonable assumptions.

20. Line 50, “whether” should change if”.

21. The research significance of intercropping is lacking in the preface, so it should be added in the preface.

22. The preamble is too concise and broad in its description and lacks a description of the advantages of intercropping.

23. Also has the crop not been studied in semi-arid environments and is it suitable for intercropping? A study between monocropping and intercropping should be done first.

Materials and methods

24. Line 58, “31.1should change “31.1 .

25. Line 66, “10℃”should change “10 ℃”.

26. Line 66,Should be deleted [8].

27. Line 68, there is no space between the number and the degree Celsius.

28. Line 71, if the title of 2.2 is incorrect, and if it would make more sense to describe it as an experimental design.

29. Line 75,fertilizer  should be deleted.

30. Line 85, please add a table that specifically describes the ratio of seeding of the two crops when intercropping.

31. Line 98, “ ml l-1 ” should change “ ml·l-1 ”.

32. Line 104. From when to measure, the specific time of each measurement, duration?

33. Add an ordinal number after each formula to find.Line 58, the growing season described here suggests describing specific months.

34. Line 107,The middle of ceptometer placed should be added  which was.

35. Line 108-109, this sentence is not very clear. It is recommended to specify the specific sampling time range.Line 122, “ g m-2 ” should change g·m-2”.

36. In 2.1, it is recommended to add a basic profile of the soil.

37. In 2.1, It is recommended to add the basic nutrient status of the land.

38. In 2.2, it is recommended that the four treatments be clearly stated.

39. It is recommended to add a planting diagram.

40. In 2.4, only two pieces of one square meter are sampled, and the number of repetitions is generally too small, at least three times.

41. Line 111, all formulas in this section do not cite references, please add them.

42. The number of test samples and the number of replicates of measurements were not specified when sampling and measuring photosynthesis, please add.

43. Line 160, there are 12 treatments described here, and the text does not specifically distinguish between the treatments, so please add them.

Results and Discussion

44. Line 168 ,to” should change for”.

45. Line 170, where did you get the relevant data for 1989-2019, it is not reflected in the table, please add it.

46. Line 183,achieve maximum light interception at 48 DAP in 2020should be changed achieve maximum light interception at 45DAP in 2020.

47. Line 185 ,37-45% ” should change 37%-45% ”.

48. Line 219223-225,g m2 mm- 1 ” should change g·m2 ·mm- 1”.

49. Line 232, There is an error in the unit corner in the vertical axis in the figure. Please check the whole text to modify it.

50. Line 233,Different letters within the same year are different at p .05should be changed Different letters within the same year are different at p 0.05.

51. Line 243,sunnhemp planted at 17, 34, 50, and 67 kg ha-1sunnhemp has only three seeding rates :16.8, 33.6, and 50.4 kg ha-1,please check and revise it carefully.

52. Line 263,There is an error in the unit corner in the vertical axis in the figure. Please check the whole text to modify it.

53. Recommendation 3.1 is more appropriate in Materials and analysis.

54. You are advised to readjust the layout of Table 1.

55. The title of 3.2 is recommended to be modified.

56. In the article p 0.05,.I suggest that p should be italicized.

57. The proposed results are written separately from the discussion.

58. In the discussion, there were too few references and only a comparison with previous research data without a deeper explanation of the mechanism.

59. Figure 2,please modify the ordinate of all figure 2.

60. Figure 2,SH,FSWhat do these two stand for respectively,please explain.

61. Figure 4a,when SH is mdand FS isno,there is a lack of salient letters.

62. Line 292-298,this paragraph should be changed to a description of figure 5,not a presentation of the diagram that is not in the text.

63. Line 314,5.1 and 5.0%” should change 5.1% and 5.0% ”.

64. Line 313-314,In 2020, CP was not different in low and medium forage sorghum treatments with 5.1 and 5.0%, respectively.If there is no difference between the two,why do different letters appear in the graph:lo:5.1 ± 0.47 band md: 5.0 ± 0.52 c .Please explain the specific reasons.

65. Line 340,according to table 2,In 2021, ADF was 39.4% with forage sorghum and 59.3% without forage sorghumshould be modified In 2021, ADF was 39.4% with forage sorghum and 58.3% without forage sorghum.

66. In the results and analysis,most of the data and conclusions of others are quoted,and the reasons for the results are not discussed and analyzed.

67. It is recommended to optimize Table 1.

68. In Figure 1, the difference between treatments is not well shown.

69. All Tables and Figures are technically very poorly prepared. The Figures should be consistent in size, font size and type etc.

70. At the end of the discussion, the limitations of this paper and the future research work should be added.

71. Abbreviations should be reduced in the text to increase the readability of the article. Please modify the whole text.

72. The layout of each figure does not make sense, putting the same set of figures on the same page.

73. There are only four treatments in Figure 1, whereas there are 12 treatments in the experimental design, why are the other treatments not depicted, please make additions.

74. What "SH" and "FS" stand for in Figure 4 is described in detail in the figure notes.

75. In the discussion, some indicators were discussed while others were not, and it was suggested that the discussion be described separately from the analysis of the results.

76. The discussion was one-sided as only a single indicator was discussed and there was no inter-indicator discussion.

Conclusions

77. The innovation of this experiment should be explained simply and clearly.

78. It is suggested to add some advantages and disadvantages compared with other studies.

79. It is suggested to supplement some limitations or deficiencies, as well as prospects for future research.

80. Line 469, the article does not have a monoculture treatment, here's how the data was obtained.

81. Line 471,474,33.7 kg seed ha-1 and 16.2 kg seed ha-1should be changed 33.6 kg seed ha-1 and 16.8kg seed ha-1.

82. Some suggestions should be added.

83. In the conclusions it is recommended to suggest which intercropping ratio is more suitable for local cultivation.

References

84. Does the DOI need to be added to the section according to the journal requirements? Are some DOI formats correct?

85. Please note that the format of references is consistent with journal requirements, and check the writing of the person’s name, journal name, and punctuation.

86. Some of the references are not in the right format, such as 1. The year font is not bold.

87. There are too few references.

Author Response

The comments that are not directly addressed below were included in the highlighted revisions of the manuscript.

  1. In the abstract, it is suggested to simplify the background, which can be shown in the introduction section. I feel that I have narrowed the background down to a bare explanation for the reason for the study.
  2. Line 12-15, this sentence is suggested to be deleted and added in the introduction. This is the objectives. Is that not relevant to include in an abstract?
  3. Line 104. From when to measure, the specific time of each measurement, duration? Measuring times were described in the methods, and if a specific time of day was required, it was also specified.
  4. Recommendation 3.1 is more appropriate in Materials and analysis. I prefer to keep 3.1 in Results because most articles include weather sections in the results rather than materials.

Round 2

Reviewer 1 Report

Comments and Suggestions for Authors

Dear Authors,

thank you for responding to some of the comments in the first round of reviews. However, the authors have provided limited responses to the questions in the first round of reviews. In my opinion, only the introduction section of the manuscript was completed sufficiently. Some of the suggestions and questions from subsequent sections were left unanswered and unaddressed. In addition, the Results and Discussion section in my opinion was slightly improved and supplemented. I suggest that the Authors provide a thorough response to the questions in the first round of reviews point by point.

In the changes made, I noticed the inconsistency of references to other Authors' works compared to the entire manuscript.

An additional suggestion that occurred to me while reviewing the manuscript to the Tables and Figures. In my opinion, they should be able to function on their own, without the text. Thus, explanations of the abbreviations used would have to be supplemented each time 

Reviewer 3 Report

Comments and Suggestions for Authors

Some of the comments included in the review were included in the revised version of the manuscript. Some comments were not taken into account. Manuscript still needs improvement.

In the abstract, we briefly describe the results of the experiment. Comparison with other studies is provided in the Discussion section.

Basic soil properties should be provided and described in the manuscript because it is an important element that may affect the results obtained. A potential reader should be able to read them. The authors' position that 'they can look for it' is inappropriate in my opinion. Moreover, you can find only some properties, i.e. granulometric composition, and it is not accurate because the ranges are given.

The manuscript should describe where the data used in the TDN calculation formula comes from, even if it is not presented in the manuscript. You may need to state that the data is unpublished or provide the source if it is published in another article. How was NDFDp determined/calculated?. There is no mention of fat determination.

It is not clear to me how there were 12 rows of sunnhemp in a plot with intercropping? I think there were 9 rows of hemp between the four rows of sorghum? I still suggest that the authors present graphically the arrangement of rows in a plot with mixed cultivation.

Another doubt. If the actual authors took into account 2 rows of sorghum and 3 sunnhemp when sampling 1 m2, they assumed a higher share of sorghum in the sampled biomass than was actually the case in the plots. Hence the low hemp biomass in intercropping. To maintain the proportions, you need to collect 1 row of sorghum and 3 rows of sunnhemp, or 2 rows of sorghum and 6 rows of sunnhemp.

Please specify what the forecrop was

Reviewer 5 Report

Comments and Suggestions for Authors

I am quite satisfied with the author's revision

Author Response

Thank you.